



# Elevation-dependent warming: observations, models, and energetic mechanisms

Michael P. Byrne[1,2], William R. Boos[3], and Shineng Hu[4]

[1]School of Earth and Environmental Sciences, University of St Andrews, St Andrews, UK
[2]Department of Physics, University of Oxford, Oxford, UK
[3]Department of Earth and Planetary Science, University of California, Berkeley, California, USA
[4]Nicholas School of the Environment, Duke University, North Carolina, USA

**Correspondence:** Michael P. Byrne (mpb20@st-andrews.ac.uk)

**Abstract.** Observational data and numerical models suggest that, under climate change, elevated and non-elevated land surfaces warm at different rates. Proposed drivers of this "elevation-dependent warming" (EDW) include surface albedo and water vapour feedbacks, the temperature dependence of longwave emission, and aerosols. Yet the relative importance of each proposed mechanism both regionally and at large scales is unclear, highlighting an incomplete physical understanding of EDW.

Here we use gridded observations, atmospheric reanalysis, and a range of climate model simulations to investigate EDW over the historical period across the tropics and subtropics (40°S to 40°N). Observations, reanalysis, and fully-coupled models exhibit annual-mean warming trends (1959–2014), binned by surface elevation, that are larger over elevated surfaces and broadly consistent across datasets. EDW varies by season, with stronger observed signals in boreal autumn and winter. Analysis of large ensembles of single-forcing simulations (1959–2005) suggests historical EDW is likely a forced response of the climate

system rather than an artefact of internal variability, and is primarily driven by increasing greenhouse gas concentrations.

To gain quantitative insight into the mechanisms contributing to large-scale EDW, a forcing/feedback framework based on top-of-atmosphere energy balance is applied to the fully-coupled models. This framework identifies the Planck and surface albedo feedbacks as being robust drivers of EDW (i.e., enhancing warming over elevated surfaces), with energy transport by the atmospheric circulation also playing an important role. In contrast, water vapour and cloud feedbacks along with weaker

radiative forcing in elevated regions oppose EDW. Implications of the results for understanding future EDW are discussed.

## 1   Introduction

Climate models and some observational studies show that, as climate warms, elevated surfaces tend to warm more rapidly than non-elevated surfaces (e.g., Yan et al., 2016; Qixiang et al., 2018). This "elevation-dependent warming" (EDW) suggests that the impacts of a changing climate will be amplified for elevated surfaces, with implications for societies and ecosystems in

mountainous regions as well as for glaciers and meltwater runoff (Bliss et al., 2014).

Amplified warming (or, more generally, differential warming) of elevated regions implies that the energetic forcing and feedback processes which control radiatively forced temperature trends (e.g., Sherwood et al., 2020) vary systematically with surface elevation. Proposed drivers of EDW based on this energetic perspective include the surface albedo feedback (Giorgi



et al., 1997), the temperature dependence of longwave emission (i.e., the Planck feedback; Pepin et al., 2015), and cloud
feedbacks (Rangwala and Miller, 2012). Radiative effects associated with increasing water vapour, in particular variations of
this feedback with surface elevation, have also been cited as a possible contributor to EDW (Rangwala et al., 2009). Some of
these proposed EDW drivers are well understood. For example, the surface albedo feedback—a positive feedback on forced
temperature changes (Hall, 2004)—is expected to be more important for high-elevation regions where surface snow and ice are
more prevalent. Similarly the negative Planck feedback is temperature dependent (Henry and Merlis, 2019; Cronin and Dutta,
2023) and expected to be weaker in colder elevated regions, thereby favouring EDW. Other factors, including radiative forcing
due to aerosols, are important for regional EDW according to some studies (Ramanathan and Carmichael, 2008; Lau et al.,
2010), yet their influence on large scales and importance relative to other EDW drivers are less clear. As-yet undiscovered
mechanisms could also influence the relative warming of elevated versus non-elevated surfaces: For example, $CO_2$ radiative
forcing is weaker in elevated regions including the Tibetan Plateau (Huang et al., 2016) but has received little attention in the
EDW literature. In summary, despite intensive research over recent decades, a comprehensive and quantitative understanding
of the physical processes driving EDW remains elusive.

In this study, we examine EDW over the historical period using gridded observations, atmospheric reanalysis, and climate
models. Our focus is on understanding the large-scale EDW signal in the tropics and subtropics (averaged from 40°S to
40°N), the consistency across observational and model datasets, and the processes influencing EDW. We focus on the tropics
and subtropics where the EDW signal is strong (Palazzi et al., 2019) and where meridional gradients in surface temperature
trends—which have the potential to complicate interpretation of the EDW signal—are relatively weak [e.g., compared to
northern mid and high latitudes where polar amplified warming manifests (Rantanen et al., 2022)]. We begin by introducing
the data and analysis techniques (section 2) before quantifying EDW using surface-air temperatures and assessing trends
across observations, models, and seasons (section 3). Using large ensembles of climate simulations, in section 4 we assess:
(i) the influence of radiative forcing versus internal variability on EDW; and (ii) the roles of specific forcing agents in driving
EDW, in particular greenhouse gases and aerosols. In section 5 we quantify and interpret the physical processes influencing
EDW using a forcing/feedback framework before finishing with a summary and conclusions (section 6).

## 2  Data and analysis

A range of monthly resolved observational and model datasets are analysed to gain insight into the historical EDW signal
and its physical drivers. On the observational side, gridded surface-air temperature anomalies from the HadCRUT5 dataset
(at $5° × 5°$ resolution; Morice et al., 2021) are analysed along with surface-air temperatures from the ERA5 reanalysis[1]
($0.25° × 0.25°$; Hersbach et al., 2020). On the model side, 20 ensemble members are analysed from each of the "all-forcing",
"all-but-greenhouse-gases", and "all-but-anthropogenic-aerosols" sets of simulations performed as part of the CESM1 Large
Ensemble (CESM1-LE; Kay et al., 2015). The latter two sets of simulations have greenhouse gases and anthropogenic aerosols,
respectively, prescribed to pre-industrial levels and are subtracted from the all-forcing runs to isolate the contributions of these

---

[1] Note that the ERA5 temperature data analysed here were accessed from the Copernicus Climate Data Store on 21-DEC-2023.



individual forcing agents to the historical temperature trends. Historical simulations from 21 fully-coupled models[2] participating in the Coupled Model Intercomparison Project Phase 6 (CMIP6; Eyring et al., 2016) are also analysed. The years used in each analysis are specified in subsequent sections, but most of our analyses use 1959–2014.

To analyse EDW, for each dataset the land-surface air temperatures (or land-surface air temperature anomalies in the case of HadCRUT5) are first binned by surface elevation. All gridboxes comprising more than 90% land are included. For ERA5 and CESM1-LE, surface elevations are derived from the surface geopotential data. Surface elevations for HadCRUT5 are obtained by regridding the ERA5 geopotential data to the HadCRUT5 grid. For CMIP6, surface elevations are taken from the GFDL-CM4 model's orography file and are regridded before being used with the other models. Eleven elevation bins are defined, with equally spaced lower bounds of 0 m surface elevation for the lowest bin and 5000 m for the highest bin; the highest bin has no upper bound and includes all gridboxes higher than 5000 m. Temperatures are averaged over each calendar year (or each calendar season) and in each elevation bin, with area weighting, prior to the multi-decadal trends being computed using ordinary least-squares regression. Binned data are plotted as a function of the mean surface elevation in each bin (e.g., Fig. 1a). Using the standard error of the slope and the t-statistic, confidence intervals are estimated for the trends under typical assumptions for ordinary least squares (e.g., normality of residuals). We repeated many of our analyses using a robust linear regression designed to be less sensitive to outliers, as implemented in the *statsmodels* robust linear models module that uses a Huber T norm with a generalised maximum likelihood method (M-estimation) to estimate the regression coefficients. Our conclusions are insensitive to this choice of regression model, but we state below any instances where slopes changed notably with the choice of statistical model.

## 3  Historical EDW on large scales

Over our tropical-subtropical region, reanalysis and gridded station data show quantitatively similar pronounced warming over elevated surfaces. Specifically, when land-surface air temperatures are binned by surface elevation and then averaged spatially and over each calendar year, as described above, linear trends over the 1959–2014 period generally increase with height (Fig. 1a). Although central estimates of the linear trends differ between the station data (HadCRUT5) and reanalysis (ERA5) in many elevation bins, the 95% confidence intervals of these trends always overlap. Spatial sampling differed substantially between the reanalysis and gridded station data, due to their different resolutions (0.25° and 5°, respectively) and some spatio-temporal gaps in the station data; one effect of this can be seen in the different mean surface elevations within each elevation bin (this is especially prominent between 3 and 4.5 km surface elevation in Fig. 1a).

The magnitude of the EDW signal, and any differences in its value between datasets, can be quantified by an "EDW index". Specifically, we define the EDW index as the slope obtained by regressing the warming trend in each elevation bin onto the mean surface elevation within each bin [this metric is similar to the "elevational gradient" analysed by Palazzi et al. (2017)].

---

[2]CMIP6 historical simulations performed by the following models are analysed: ACCESS-CM2, AWI-ESM-1-1-LR, BCC-ESM1, CESM2-FV2, CESM2-WACCM-FV2, CanESM5, FGOALS-g3, GFDL-CM4, GFDL-ESM4, INM-CM4-8, INM-CM5-0, IPSL-CM5A2-INCA, IPSL-CM6A-LR, IPSL-CM6A-LR-INCA, KACE-1-0-G, KIOST-ESM, MIROC6, MPI-ESM-1-2-HAM, MPI-ESM1-2-LR, MRI-ESM2-0, and NorESM2-LM.




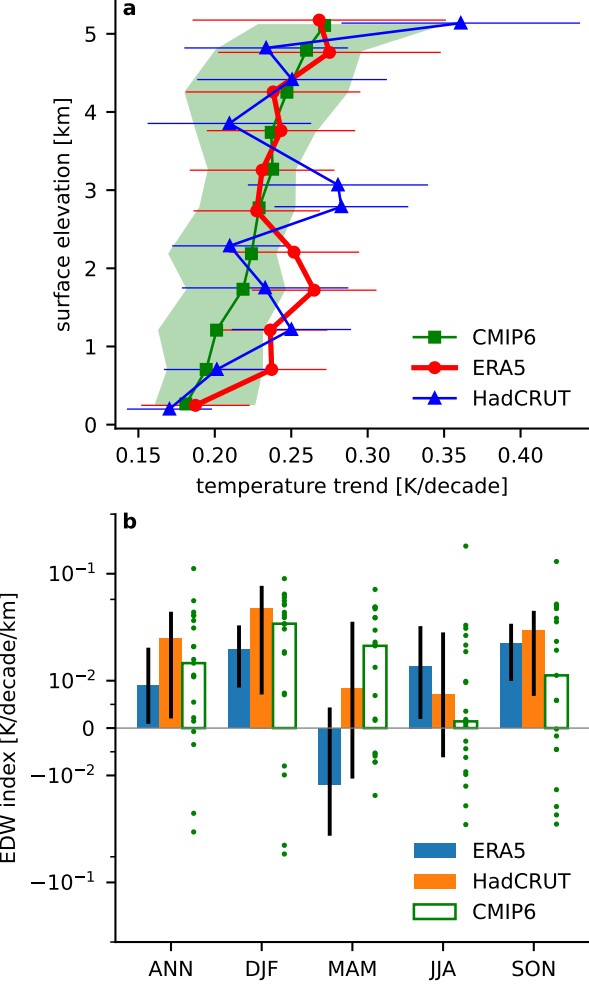

**Figure 1.** (a) Land surface-air temperature trends binned by surface elevation for the ERA5 reanalysis, HadCRUT5 dataset, and CMIP6 historical simulations (1959–2014, data averaged from 40°S to 40°N). Here and in subsequent figures, the trends are plotted as a function of the mean surface elevation in each bin. For ERA5 and HadCRUT5, error bars are the 95% confidence intervals. For the CMIP6 simulations, the line with squares shows the median temperature trend among the models in each elevation bin, and shading shows the interquartile range. (b) Elevation-dependent warming (EDW) index [i.e., the inverse of the slope of the curves in (a), as described in the text] for ERA5, HadCRUT5, and CMIP6 computed for the annual mean and for each season. Error bars for ERA5 and HadCRUT5 indicate the 95% confidence intervals; for CMIP6, the open bar shows the median EDW index among the models, and dots show the EDW index in each model. The vertical axis in (b) has a "symmetrical logarithmic scale" that is linear between $\pm0.02$ K/decade/km and logarithmic beyond that range in positive and negative directions (this scale is used because a few model outliers have EDW indices much larger than observed).

This yields, for example, EDW indices of $0.0089 \pm 0.0080$ K/decade/km for ERA5 and $0.0189 \pm 0.0169$ K/decade/km for HadCRUT5 (Fig. 1b; the uncertainties listed here correspond to 95% confidence intervals). These values are not statistically



distinct from each other, and both have confidence intervals that do not include zero (the p-values are 0.043 and 0.042, re-
spectively). Using a robust linear model instead of ordinary least squares yields slightly smaller central estimates of the EDW

indices, 0.0087 K/decade/km for ERA5 and 0.0180 for HadCRUT5, with respective p-values of 0.058 and 0.035.

All of the above results were for annual mean temperatures, and the EDW indices vary by season (Fig. 1b). Boreal autumn
(SON) and winter (DJF) values are larger than spring (MAM) and summer (JJA) values in both datasets; in HadCRUT5 only
the autumn and winter seasons are statistically distinct from zero at the 95% level. The larger EDW signal in cool seasons will
turn out to be consistent with the larger contribution to EDW of mechanisms that are stronger at colder temperatures (i.e., the

Planck and surface albedo feedbacks, discussed in section 5 below).

On examining the historical CMIP6 simulations, we find that the ensemble median EDW indices and warming trends in
each elevation bin are roughly similar to those of our two observational datasets (Fig. 1). For the warming trends, error bars
for the two observational datasets fall within the interquartile range of the CMIP6 models in every elevation bin. The CMIP6
median EDW index falls within the error bars of the two observational datasets for annual mean data, although the model

ensemble spans a large range with four models exhibiting reduced warming over elevated surfaces (negative EDW index), and
a few models having EDW indices that are roughly an order of magnitude larger than observed. Like observations, the models
exhibit larger EDW in cool seasons, although the largest values are achieved in boreal winter and spring rather than in the
observed seasons of winter and autumn. Given the large range of EDW indices across the models, this difference between
spring and autumn may not be especially meaningful in the models.

Do these measures of differential warming have geographic correspondence between the models and our two observational
datasets? Annual mean warming trends (over the same 1959–2014 period used above) are largest over many of the same
orographic regions in all three datasets: the Tibetan and Iranian Plateaus, the North American Cordillera, and the Brazilian
Highlands in eastern South America (Fig. 2). Warming is also strong over the Arabian Peninsula and Sahara in all datasets;
although parts of these regions contain high orography, there does not seem to be a strong relation between the warming

rate and surface elevation over Africa and the Arabian Peninsula. Since factors other than surface elevation are expected to
influence the warming rate, such as surface aridity (Byrne and O'Gorman, 2013), we do not expect the map of warming rate to
have the same pattern as the map of surface elevation. Given the prominence of enhanced warming over off-equatorial regions,
particularly in the northern hemisphere (Fig. 2), it seems worthwhile to assess whether the EDW signal seen in Fig. 1 might be
an artefact of the polar amplification of warming that is seen primarily in the northern hemisphere (e.g., Pithan and Mauritsen,

2014). We assess this possibility in an Appendix, showing that the association of latitude with warming is insufficiently large
to explain the majority of the observed EDW signal.

## 4 Drivers of EDW: internal variability versus radiative forcing

Is the historical EDW described in section 3 a forced response of the climate system (e.g., to increasing greenhouse gases)? Or
is it potentially an artefact of internal variability? To address these questions, we analyse data from the CESM1-LE simulations

(1959–2005) to isolate the relative contributions of external radiative forcing and natural internal variability to historical EDW.





**Figure 2.** Spatial distribution of linear temporal trends of annual mean surface air temperature between 1959–2014 in (a) ERA5, (b) Had-CRUT5, and (c) CMIP6, all in $\mathrm{K/decade}$, with the CMIP6 plot showing the median trend across the 21-model ensemble. Stippling marks regions where the 95th-percentile confidence interval includes zero in (a) and (b), and where the interquartile range across the model ensemble includes zero in (c). White regions in (b) lack data. Trends over ocean are shown for ERA5 and CMIP6 for reference, but are not included in any of our EDW analyses.

Across the all-forcing ensemble, 18 out of 20 members show a positive EDW index (i.e., enhanced warming at elevation; Fig. 3), implying that historical EDW is very likely, at least in part, to be radiatively forced. The EDW index varies substantially across members in the all-forcing ensemble, from +0.0287 to -0.0082 $\mathrm{K/decade/km}$ (Fig. 3b). This suggests an important role for internal variability in affecting the magnitude of the historical EDW signal, consistent with Palazzi et al. (2019). The EDW



indices from the HadCRUT5 and ERA5 datasets are $0.0140 \pm 0.0189$ K/decade/km and $0.0099 \pm 0.0096$ K/decade/km, respectively (Fig. 3b), which are similar to the ensemble-mean all-forcing EDW index (0.0135 K/decade/km) and fall within the ensemble spread (these HadCRUT5 and ERA5 values differ from those given in the previous section because here we use an analysis period ending in 2005). These results suggest that both radiative forcing and internal variability have played an important role in shaping historical EDW.

Greenhouse gases and anthropogenic aerosols, two important radiative forcing agents (Smith et al., 2020), can both drive regional patterns of surface temperature change (Mitchell et al., 1995). To advance understanding of EDW, it is important to assess which forcing agent is responsible for the large-scale EDW signal over the historical period. To that end, we analyse the CESM1-LE single-forcing simulations, wherein greenhouse gases or aerosols are prescribed to pre-industrial levels so as to isolate the contributions of these forcing agents to historical trends (see section 2). We find that greenhouse gases are

the dominant driver of historical EDW (cf. grey and red markers in Fig. 3b). Aerosol forcing has only a weak influence on large-scale EDW (blue markers in Fig. 3b), but could potentially be important on regional scales (e.g., Pepin et al., 2015). Inter-member correlations between tropical mean temperature trends and the EDW index are weak, for both the all-forcing and single-forcing experiments (Fig. 3b), suggesting that the internal variability influencing tropical mean warming is different in character from the variability controlling EDW, and that the magnitude of EDW is not simply determined by the rate of overall

tropical warming.

## 5 Processes influencing EDW in historical simulations

### 5.1 Forcing/feedback framework

In this section, we investigate the physical drivers of EDW in the CMIP6 historical simulations. To decompose the processes influencing annual mean surface air warming at different elevations, we start by considering atmospheric energy balance in a

forcing/feedback framework. At steady state, the local atmospheric energy budget can be written as:

$$0 = R_t - R_s - \nabla \cdot F_a, \tag{1}$$

where $R_t$ is the net radiative flux at top-of-atmosphere (TOA) and $R_s$ is the net energy flux between the atmosphere and surface (radiative plus turbulent fluxes). Both $R_t$ and $R_s$ are defined as positive downwards. $\nabla \cdot F_a$ is the divergence of the horizontal moist static energy (MSE) flux, $F_a$, integrated over the depth of the atmosphere. The heat capacity of the land surface is

relatively small, so on annual and longer timescales the fluxes into and out of the land surface are expected to be approximately balanced (e.g., see Fig. 9 in Liu et al., 2017). We therefore neglect the surface flux term in (1) to give:

$$0 \approx R_t - \nabla \cdot F_a. \tag{2}$$

Equation (2) implies a tight coupling between TOA radiative fluxes and atmospheric energy transport over land. Trends in radiative fluxes and atmospheric energy transport, for example in response to global warming, are also tightly coupled:

$$0 \approx \delta R_t - \nabla \cdot \delta F_a, \tag{3}$$



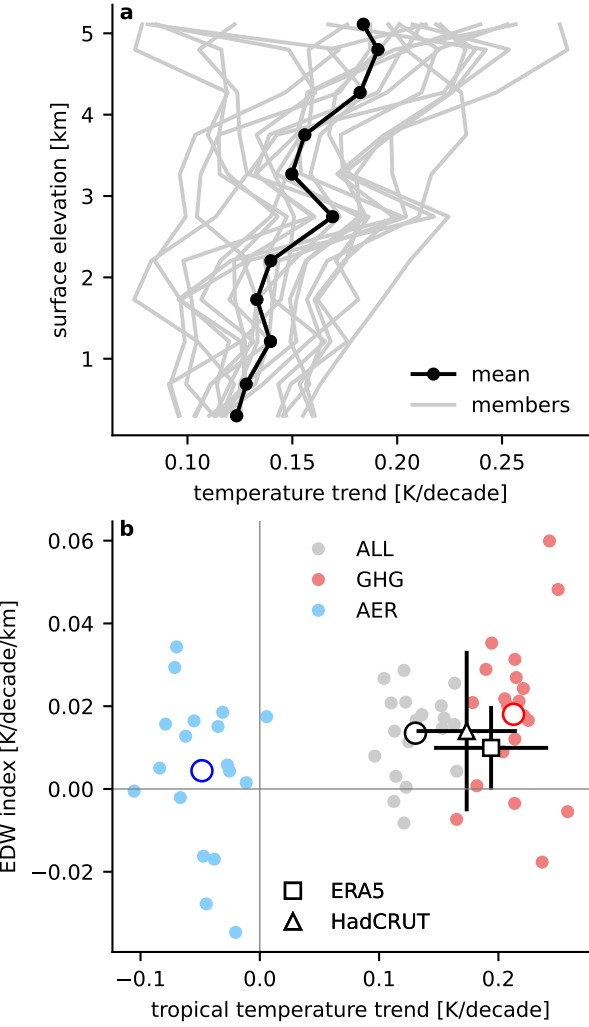

**Figure 3.** (a) Surface air temperature trends binned by surface elevation for 20 ensemble members (grey lines) and the ensemble mean (black line with dots) from the CESM1-LE all-forcing simulations (1959–2005). (b) Scatterplot of the EDW index versus surface air temperature trend averaged over tropical land (40°S to 40°N) for each ensemble member in the all-forcing simulations (ALL; grey dots) and in the cases where only greenhouse gas forcing (GHG; red dots) and only anthropogenic aerosol forcing (AER; blue dots) are changing over the historical period. The large dots indicate the ensemble means for the ALL, GHG, and AER cases. Corresponding values for the ERA5 reanalysis (black square) and HadCRUT5 observations (black triangle) are also shown, with error bars indicating the 95% confidence intervals.

where $\delta$ denotes a linear trend.





As is standard in physical climate science (e.g., Hansen et al., 1984), we next express trends in the net TOA radiative flux as a linear sum of a radiative forcing, $F$, and a temperature-mediated feedback term:

$$\delta R_t = F - \lambda \delta T_s, \tag{4}$$

where $\lambda$ is the climate feedback parameter (Gregory et al., 2004) and $T_s$ is the surface air temperature. The $\lambda$ parameter is composed of a variety of individual feedback processes that are assumed to be independent of one another:

$$\lambda = \lambda_{PL} + \lambda_{LR} + \lambda_{WV} + \lambda_{CL} + \lambda_{AL} + \lambda_{ST}, \tag{5}$$

where the subscripts "PL", "LR", "WV", "CL", "AL", and "ST" denote the Planck, lapse rate, water vapour, cloud, surface albedo, and stratospheric feedbacks, respectively. Substituting (5) into (4), then (4) into (3) and rearranging, we obtain:

$$0 = F - (\lambda_{PL} + \lambda_{LR} + \lambda_{WV} + \lambda_{CL} + \lambda_{AL} + \lambda_{ST}) \delta T_s - \nabla \cdot \delta F_a, \tag{6}$$

where the approximation symbol associated with (3) has been dropped.

Equation (6) is the basis for the framework we employ to decompose and quantify the processes contributing to EDW. In particular, following Goosse et al. (2018), we split the total surface air temperature trend into components associated with different processes. To do this we first define $\overline{\lambda_{PL}}$ to be the global mean Planck feedback, with $\lambda'_{PL}$ denoting a local departure 170 from this global mean. Inserting $\lambda_{PL} = \overline{\lambda_{PL}} + \lambda'_{PL}$ into (6) and rearranging we find:

$$\delta T_s = (F - [\lambda'_{PL} + \Sigma_i \lambda_i] \delta T_s - \nabla \cdot \delta F_a) \times (1/\overline{\lambda_{PL}}), \tag{7}$$

where $i = [LR, WV, CL, AL, ST]$. Each term on the right-hand side of (7) represents a contribution, from a particular process, to the local surface air temperature trend. Through analysing how these contributions vary with surface elevation, we aim to quantify and gain physical insight into the processes shaping EDW.

**5.2 Methodology**

The processes driving EDW are quantified in fully-coupled CMIP6 simulations (see section 2 for the list of models). In particular, we analyse trends in surface air temperature in the historical simulations (1959–2014) and assess how these trends vary with surface elevation.

To quantify the energetic contributions to the temperature trends, we need to estimate the radiative forcing, radiative feed-180 backs, and atmospheric MSE transport [see (7)]. The TOA radiative flux trends associated with the Planck, lapse rate, water vapour, and surface albedo feedbacks are computed by convolving radiative kernels with trends in tropospheric climate variables (i.e., temperature, specific humidity, and surface albedo) (Soden and Held, 2006). The flux trends are normalised by the local surface air temperature trends to convert into local feedbacks (with units $\mathrm{W/m^2/K}$). To estimate the various feedbacks, we use the monthly resolved Geophysical Fluid Dynamics Laboratory radiative kernels (Soden et al., 2008). Cloud feedbacks 185 are computed by adjusting trends in the TOA cloud radiative effect to account for cloud masking effects (Soden et al., 2008). In these calculations and similar to Soden and Held (2006), the tropopause is specified to be at 100 hPa at the equator and





varies linearly with increasing absolute latitude to 300 hPa at the poles. The stratospheric feedback is computed by convolving trends in temperature and specific humidity above the tropopause with the temperature and humidity radiative kernels. Note that the stratospheric contribution to TOA flux trends is often considered as an "adjustment" to radiative forcing rather than a

temperature-mediated feedback (Sherwood et al., 2015).

Radiative forcing is estimated as a residual from the TOA energy budget (4), by subtracting from the total radiative flux trend contributions due to the various feedback processes (this is an estimate of the "instantaneous radiative forcing"; IRF). Following Kramer et al. (2021), we use a cloud masking constant of 1.24 to convert from a clear-sky IRF to an all-sky IRF. The trend in atmospheric MSE divergence over land, $\nabla \cdot \delta F_a$, is approximated from (3) as the trend in net TOA radiative flux,

thereby neglecting trends in surface and atmospheric energy storage.

Below, we apply this methodology to investigate temperature trends as a function of surface elevation. This complements previous work using similar frameworks to understand the drivers of polar warming (Pithan and Mauritsen, 2014; Hahn et al., 2020) and the land-ocean warming contrast (Toda et al., 2021), and aims to directly quantify how a range of physical processes contribute to EDW.

### 5.3    Contributions to EDW

The CMIP6 historical simulations show an amplified warming over elevated surfaces (Fig. 4) that is broadly consistent with HadCRUT5 and ERA5 data (Fig. 1). The multi-model median surface air temperature trend averaged over the two highest elevation bins is 42% larger than the average trend for the two lowest bins (0.2659 vs 0.1877 K/decade), and the multi-model median EDW index is 0.0137 K/decade/km (Fig. 4b). Below we quantify and discuss the energetic processes influencing this

historical EDW signal.

#### 5.3.1    Local Planck feedback

The Planck feedback quantifies the sensitivity of blackbody emission to a change in temperature and is a negative feedback, suppressing the temperature response to external forcing (Knutti and Rugenstein, 2015). Following the Stefan-Boltzmann law, the Planck feedback is temperature dependent with a magnitude approximately proportional to $T_s^3$ (e.g., Hartmann, 2016).

The cooling effect of the Planck feedback is therefore expected to be weaker for colder, high-elevation surfaces compared to warmer, low-elevation surfaces. This implies that the local Planck feedback contribution to the temperature trends favours amplified warming over elevated surfaces (Fig. 4a).

Previous studies have discussed the Planck feedback as a potential driver of EDW (e.g., Pepin et al., 2015). Here, we quantify how this mechanism influences EDW and interpret its sign and magnitude using simple physical arguments. The strength of

the cooling associated with the local Planck feedback scales with the ratio of the local anomaly to the global feedback [i.e., $-(\lambda'_{PL}/\overline{\lambda_{PL}}) \times \delta T_s$; see (7)]. The local feedback anomaly, defined as $\lambda'_{PL} = \lambda_{PL} - \overline{\lambda_{PL}}$, is proportional to the difference between the cubes of the climatological local and global mean temperatures, i.e. $\lambda'_{PL} \propto T_s^3 - \overline{T_s}^3$ where $\overline{T_s}$ is the global mean temperature. Consequently, the effect of the local Planck feedback on EDW is a simple function of climatological temperature and scales as $-(\lambda'_{PL}/\overline{\lambda_{PL}}) = 1 - (T_s/\overline{T_s})^3$. For cold regions, where $T_s < \overline{T_s}$ and $1 - (T_s/\overline{T_s})^3 > 0$, the simple scaling suggests



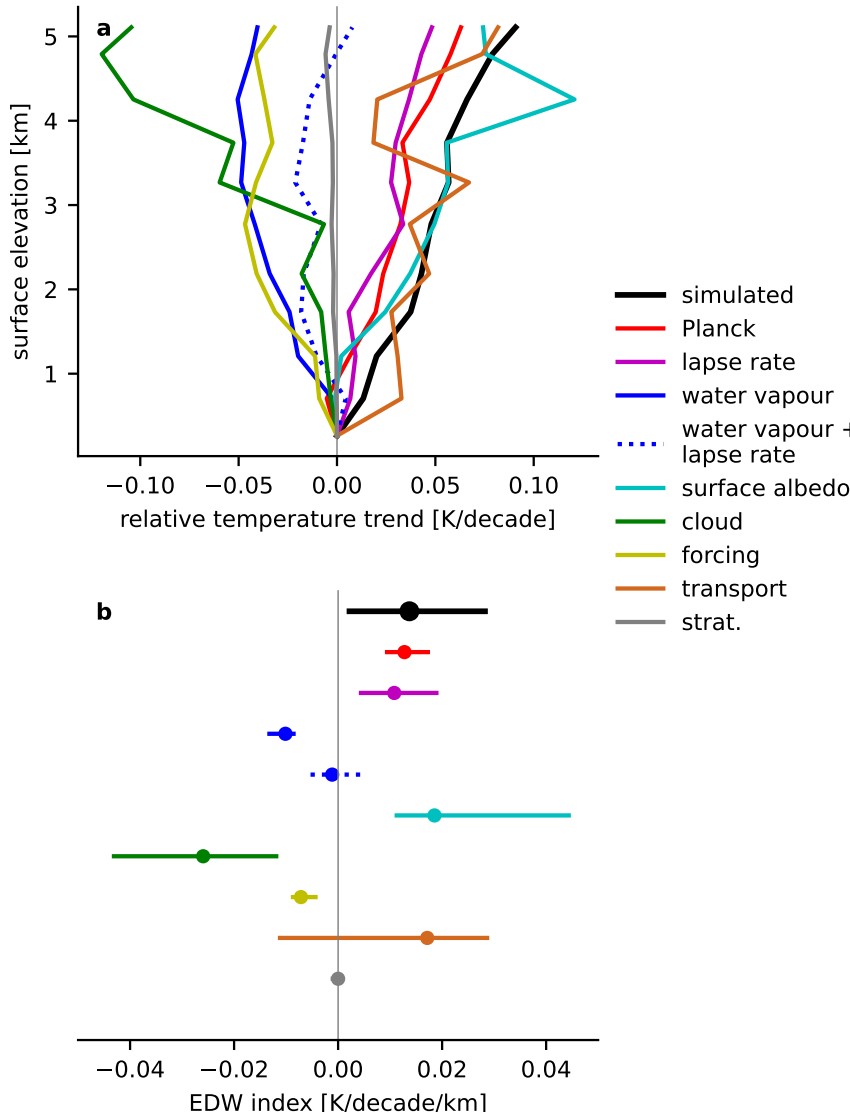

**Figure 4.** (a) Multi-model median surface air warming trends binned by surface elevation for the CMIP6 historical simulations (black line). Trends are computed over 1959–2014 and only land gridboxes between 40°S and 40°N are included. Components of the warming trends associated with different energetic processes, following (7), are also shown (coloured lines). Note that the warming trends relative to the trend for the lowest bin are plotted so as to highlight variations with surface elevation. (b) Simulated EDW index (black) computed for the CMIP6 simulations along with the contributions from individual processes (colours). Note that a positive EDW index indicates an increasing temperature trend with surface elevation. Dots show the multi-model median values and lines show the interquartile ranges.

that the local Planck feedback has a warming influence on temperature trends. But for warm regions, where $T_s > \overline{T_s}$, the local Planck feedback has a cooling influence. This temperature dependence explains why the local Planck feedback enhances





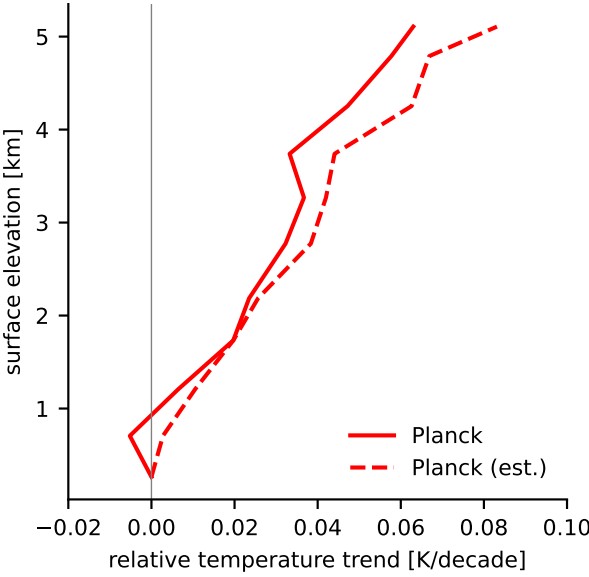

**Figure 5.** Multi-model median Planck feedback contribution to the surface air temperature trends binned by surface elevation for the CMIP6 historical simulations (solid red line). The dashed red line shows a simple estimate of the Planck component based on the variation of climatological temperature with surface elevation (see section 5.3.1 for details).

warming of cold, high-elevation surfaces relative to warm, low-elevation surfaces (Fig. 4a). Our simple estimate of the influence of the local Planck feedback on EDW is consistent with simulations (Fig. 5), suggesting that basic physics—namely the temperature dependence of blackbody emission—has a robust strengthening influence on EDW across models (Fig. 4b). The

climatological temperature gradient between low-elevation and high-elevation surfaces is larger in boreal winter compared to the annual mean (not shown), suggesting that the influence of the Planck feedback on EDW is stronger when temperatures are cold, which likely contributes to the large observed EDW signal in DJF (Fig. 1b).

### 5.3.2 Lapse rate feedback

Like the local Planck feedback, the lapse rate feedback also contributes to amplified warming of elevated surfaces (Fig. 4).

Changes in the vertical temperature gradient (i.e., the lapse rate) affect the efficiency by which the atmosphere cools radiatively to space (Colman and Soden, 2021), resulting in a temperature-mediated feedback. This feedback is negative in the tropics and subtropics (Linke et al., 2023), where amplified warming in the middle troposphere due to increasing water vapour and latent heat release enhances the atmosphere's radiative cooling efficiency. But this negative feedback is weaker over elevated surfaces where lapse rate changes are weaker, consistent with the atmosphere being colder and drier (Randel et al., 1996; Joshi et al.,

235 2008).




### 5.3.3 Water vapour feedback

Closely connected to the lapse rate feedback is the water vapour feedback (Lambert and Taylor, 2014), which robustly opposes EDW across models (Fig. 4). The water vapour feedback is the strongest positive feedback in the climate system (Soden and Held, 2006), amplifying the temperature response by increasing atmospheric absorption of longwave and shortwave radiation (Manabe and Wetherald, 1967). But the water vapour feedback is less positive for elevated surfaces and, therefore, acts to oppose EDW (Fig. 4).


The atmosphere is thinner and drier in high-elevation regions (e.g., the Tibetan Plateau; Randel et al., 1996). Therefore, absent large changes in relative humidity in a warming climate, trends in column integrated water vapour—and the water vapour feedback (Held and Soden, 2000)—are expected to be weaker for high-elevation versus low-elevation regions. Our finding, based on radiative kernel calculations, that the water vapour feedback opposes EDW contrasts with previous studies which argue, for example based on statistical relationships between humidity and surface downwelling radiation (Rangwala et al., 2010), that increases in water vapour favour EDW.


The strong and well-understood coupling between changes in water vapour and lapse rates results in the two feedbacks often being considered together (Colman, 2003). Following this precedent, we assess the combined influence of the water vapour and lapse rate feedbacks on EDW and find it to be weak (Fig. 4a) and not robust in terms of sign across models (Fig. 4b).


### 5.3.4 Surface albedo feedback

The surface albedo feedback is positive and strengthens with elevation, thereby strongly contributing to EDW (Fig. 4a). The link between surface albedo feedback and EDW is intuitive: at elevation, particularly in tropical/subtropical regions, there is typically more snow and ice to melt making the surface albedo more sensitive to warming. Although the role of this feedback in driving EDW is well established (Giorgi et al., 1997; Minder et al., 2018), here we quantify its effect at large scales and place its influence on EDW in the context of other mechanisms. The spread across models in the surface albedo component of EDW is substantial (Fig. 4b), suggesting that improved observations and modelling of surface snow and ice processes are important for constraining EDW. The strong influence of the surface albedo feedback on EDW suggests that the large observed signal in boreal winter (Fig. 1b) is potentially related to trends in surface albedo, which might be expected to be stronger in colder seasons where snow and ice are more prevalent.



### 5.3.5 Cloud feedbacks

Radiative feedbacks associated with clouds strongly oppose EDW, particularly in high-elevation regions (Fig. 4a). This relative cooling influence on elevated surfaces is robust in sign across models (Fig. 4b), and is driven primarily by longwave cloud effects for surface elevations below approximately 3.5 km and by shortwave effects higher up (Fig. 6). This result, demonstrating that clouds exert a relative cooling effect on elevated regions in a warming climate, contrasts with previous work suggesting that cloud radiative effects contribute to EDW (Liu et al., 2009).






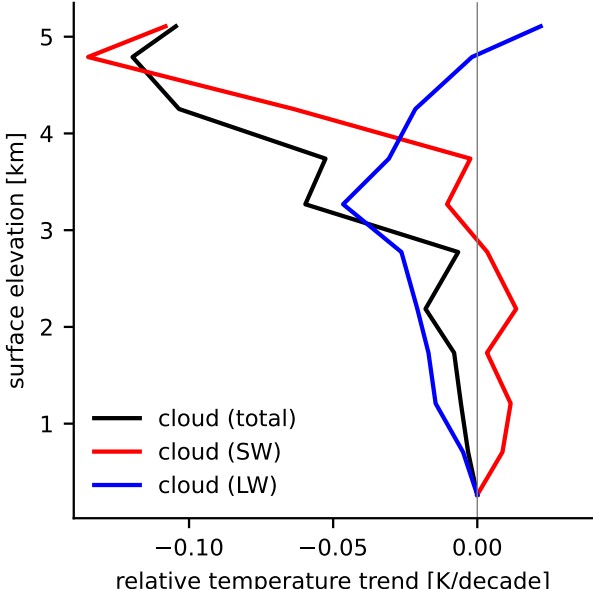

**Figure 6.** Multi-model median cloud feedback contribution to surface air temperature trends binned by elevation for the CMIP6 historical simulations (black line). The individual contributions from shortwave (SW) and longwave (LW) cloud feedbacks are also shown (red and blue lines, respectively).

The longwave cloud feedback over tropical land is typically negative in climate models (Kamae et al., 2016), with the feedback generally more negative for elevated surfaces (Fig. 6). Cloud feedbacks over land have received relatively little attention in the literature, perhaps due to their small magnitude (Sherwood et al., 2020), but have been linked to decreases

in cloud amount associated with decreases in land relative humidity (Kamae et al., 2016; Sherwood et al., 2020). The more negative longwave cloud feedback over elevated surfaces could be due to stronger decreases in cloud amount [as shown by (Liu et al., 2009)] or potentially due to changes in cloud altitude. The tropical mean land shortwave cloud feedback is positive in models, largely due to decreases in high cloud amount (Kamae et al., 2016). But this shortwave feedback is negative above surface elevations of approximately 4 km, leading to an important cooling influence on high-elevation temperature trends (Figs.

4a and 6). Detailed study of how clouds in elevated regions respond to warming is a priority for future work, given the strong yet uncertain influence of cloud feedbacks on EDW (Fig. 4b).

### 5.3.6 Stratospheric feedback

The influence on EDW of stratospheric feedbacks associated with temperature and humidity trends is negligible (Fig. 4) and is not discussed further.





### 5.3.7 Radiative forcing

Radiative forcing varies from region to region, even in response to spatially uniform changes in forcing agents (Huang et al., 2016). For example, atmospheric $CO_2$ dominates radiative forcing over the historical period (IPCC, 2023), but this forcing— both for all-sky and clear-sky conditions—is smaller in polar regions and over elevated surfaces (e.g., the Tibetan Plateau; Huang et al., 2017). Weaker radiative forcing at higher surface elevations opposes EDW[3] (Fig. 4a) and is consistent with a recent theory suggesting that clear-sky $CO_2$ forcing depends on the temperature difference between the surface and stratosphere (Jeevanjee et al., 2021). Colder surface temperatures therefore contribute to $CO_2$ forcing being weaker in elevated regions, explaining why the spatial pattern of radiative forcing opposes EDW (Fig. 4b). Forcing due to aerosols is more spatially inhomogeneous compared to $CO_2$ forcing (Shindell et al., 2013) and is potentially important for driving regional EDW signals. But in our CESM1-LE analysis aerosol forcing made a weak, at best, contribution to large-scale EDW (Fig. 3b).

### 5.3.8 Transport term

MSE transport by the atmospheric circulation contributes strongly to the multi-model median EDW signal by preferentially warming elevated regions (Fig. 4a), though there is considerable spread across models (Fig. 4b). In the climatological mean, relative to low-elevation regions there is anomalous convergence of MSE by the atmosphere over high-elevation regions (Fig. 7a), prior to the imposition of a radiative forcing. The strength of this anomalous MSE convergence increases as climate warms (Fig. 7b), contributing to amplified warming of elevated surfaces.

To interpret why anomalous MSE convergence over elevated regions strengthens in a warming climate, we begin by assuming that the atmosphere diffuses MSE downgradient (Flannery, 1984; Rose et al., 2014):

$$F_a \approx -\mathcal{D}\nabla h, \tag{8}$$

where $\mathcal{D}$ is the diffusivity and $h$ is the surface air MSE. Applying the convergence operator to (8) and neglecting spatial structure in the diffusivity, we obtain: $-\nabla \cdot F_a \approx \mathcal{D}\nabla^2 h$. Assuming constant diffusivity, trends in MSE convergence can be approximated as: $-\nabla \cdot \delta F_a \approx \mathcal{D}\nabla^2 \delta h$. Combining these diffusive relationships and rearranging, we find:

$$-\nabla \cdot \delta F_a \approx -\nabla \cdot F_a \overline{\left(\frac{\nabla^2 \delta h}{\nabla^2 h}\right)}. \tag{9}$$

Equation (9) suggests that trends in atmospheric MSE convergence are proportional to: (i) the climatological convergence ($-\nabla \cdot F_a$); and (ii) the ratio of the Laplacian of MSE trends to the Laplacian of climatological MSE. To dampen local-scale noise associated with the Laplacian operator, the ratio of Laplacian terms is averaged over all land (from 40°S to 40°N) when evaluating (9) [area averaging is denoted by an overbar].

This diffusive scaling provides a reasonable estimate of the variation with elevation of convergence trends (Fig. 7b). Given the Laplacian term is averaged over all land and therefore has no explicit dependence on surface elevation, Figure 7b suggests

---

[3]In our analysis, historical forcing in the CMIP6 simulations is estimated as an instantaneous radiative forcing (IRF; see section 5.2). Computing an effective radiative forcing (ERF) for a single model (GFDL-CM4) using fixed-SST simulations from the Radiative Forcing Model Intercomparison Project (RFMIP; Pincus et al., 2016), we find that the influence on EDW is similar compared to the IRF method (Fig. S1).





that the multi-model median trend in anomalous convergence over elevated regions—and hence the contribution of atmospheric

MSE transport to EDW—is broadly explained by the climatological convergence[4]. This diffusive argument suggests that trends in convergence over elevated regions are, approximately, driven by the climatological structure of MSE transport: Because there is anomalous atmospheric convergence over elevated regions in the climatological mean (Fig. 7a), this convergence strengthens in a changing climate and thereby contributes to amplified warming over elevated regions. Whether the climatological convergence pattern is strengthened or weakened in a warming climate depends on the sign of the Laplacian ratio [see (9)], which is

positive for the multi-model median suggesting that the spatial structure in surface air MSE becomes more pronounced over tropical land as climate warms. Understanding this ratio in more detail is a topic for future work.

## 6    Summary and conclusions

EDW has been studied for over two decades yet debate persists on the physical processes driving this phenomenon and its robustness across datasets. In this study, we examine historical EDW using gridded observations, reanalysis data, and climate

models. Averaged over the tropics and subtropics, positive annual mean EDW indices (i.e., larger surface air warming trends for high-elevation regions) are identified in HadCRUT5 observations, ERA5 reanalysis, and across the CESM1-LE and CMIP6 ensembles. The EDW index varies substantially across seasons, with boreal winter (DJF) showing the strongest relative warming of high-elevation surfaces. The warming trends binned by surface elevation are reasonably consistent across the datasets, suggesting that EDW is a robust response of the climate system to historical warming that is broadly captured by climate mod-

els. A simple calculation is used to argue that the majority of the EDW signal cannot be explained by elevated regions being situated further poleward where the warming trend is larger. Rather, EDW appears to be a phenomenon that is at least partially distinct from polar amplified warming.

Two approaches are taken to understand the mechanisms controlling annual mean EDW. First, analysis of twenty ensemble members from the CESM1-LE indicates—consistent with previous work—that historical EDW is likely to be a radiatively

forced response of the climate system and not an artefact of internal variability. Furthermore, internal multi-decadal variability produces uncorrelated changes in the magnitude of EDW and the tropics-wide temperature trend; in other words, the magnitude of EDW is not simply set by the rate of overall tropical warming. Single-forcing CESM1-LE simulations also demonstrate that EDW, at least on large scales, is primarily driven by radiative forcing associated with increasing greenhouse gas concentrations rather than anthropogenic aerosols.

Second, a forcing/feedback framework based on TOA energy balance is used to directly quantify the physical processes contributing to, and opposing, EDW in the historical CMIP6 simulations. Consistent with previous studies, the surface albedo and Planck feedbacks favour amplified warming of elevated surfaces. The analysis also demonstrates that the (positive) water

---

[4] An alternative diffusive scaling, which utilises similar assumptions to those used to derive (9), is: $-\nabla \cdot \delta F_a \approx (-\nabla \cdot F_a/\nabla^2 h) \times \overline{\nabla^2 \delta h} \approx \mathcal{D} \times \overline{\nabla^2 \delta h}$. This alternative scaling was tested in the CMIP6 simulations by first diagnosing the diffusivity using $\mathcal{D} \approx -\nabla \cdot F_a/\nabla^2 h$ and then combining with the Laplacian of the MSE trends averaged over all land from $40°$S to $40°$N (i.e., $\overline{\nabla^2 \delta h}$). This alternative scaling does not capture the variation with elevation of simulated trends in MSE convergence (not shown), suggesting that the elevation dependence of diffusivity is not the primary driver of the elevation dependence of atmospheric MSE convergence trends (Fig. 7b).



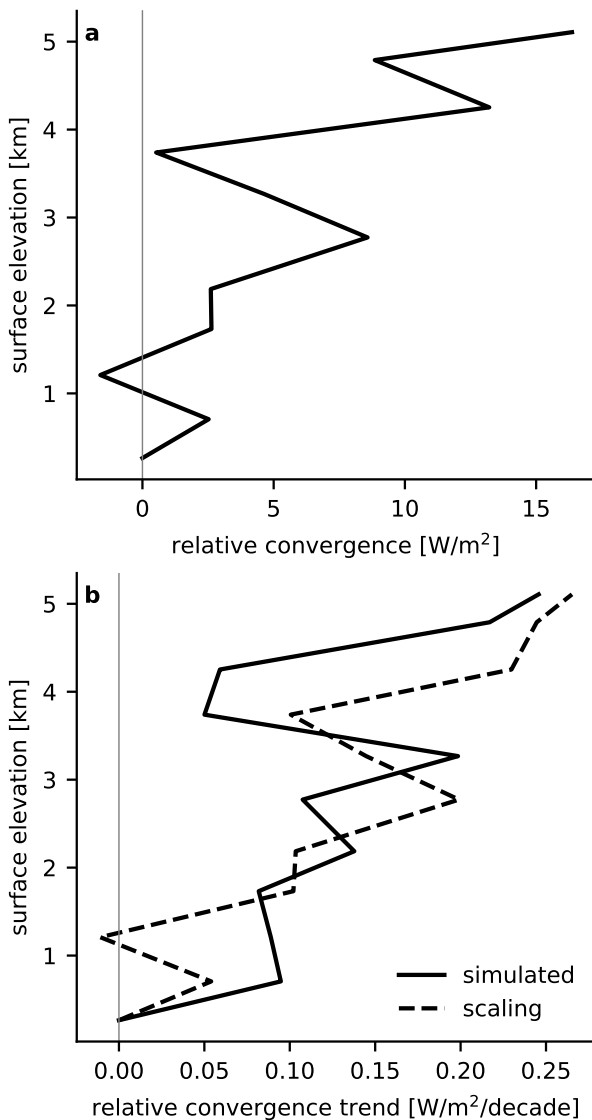

**Figure 7.** Multi-model median (a) climatological mean atmospheric convergence of moist static energy ($-\nabla \cdot F_a$) and (b) trends in convergence (solid black line) for the CMIP6 historical simulations. Both quantities are plotted relative to their values in the lowest elevation bin. The dashed black line in panel (b) shows a diffusive scaling for the convergence trend [see (9)].

vapour feedback is weaker for high-elevation regions and therefore opposes EDW. This result contrasts with other studies, which argue [e.g., based on statistical relationships between water vapour and downwelling longwave fluxes in models (Rangwala et al., 2010)] that the radiative effects of water vapour favour EDW. Here we use radiative kernels to isolate the influence of water vapour on EDW; this difference in methodology compared to previous studies may explain the differing results.



The effect of the water vapour feedback on EDW is largely cancelled by the lapse rate feedback, as expected from physical reasoning. Cloud feedbacks are shown to strongly oppose EDW, a result which also contrasts with previous work. The forcing/feedback framework reveals two additional contributors to EDW that have received little attention to date. The first is
radiative forcing, which is shown to oppose EDW because it is weaker (i.e., less positive) over cold, elevated surfaces. The second is energy transport by the atmospheric circulation, which favours EDW in the majority of CMIP6 models—a result which can be interpreted using diffusive arguments—but exhibits considerable inter-model uncertainty.

The analyses presented here provide new, quantitative insights into the processes driving EDW. Future research could expand this work by examining large-scale EDW signals across different large ensemble projects, different reanalysis products, and
different observational datasets. The forcing/feedback framework could also be extended to examine the processes controlling EDW in specific seasons, specific regions, beyond 40°N and 40°S, and in simulations of both past and future climate states. The results in this study suggest that uncertainty in future EDW may be driven primarily by uncertainties in how atmospheric energy transport over land responds to climate change, along with uncertainties in surface albedo and cloud feedbacks. Improved understanding of how these processes vary with surface elevation is essential for building reliable EDW projections, with
benefits for communities living in mountainous regions.

*Code and data availability.* ERA5 data are available from the Copernicus Climate Data Store (https://cds.climate.copernicus.eu) and Had-CRUT5 data are available through the Met Office Hadley Centre (https://www.metoffice.gov.uk/hadobs/hadcrut5/). CESM1-LE data are available through the National Center for Atmospheric Research (https://www.cesm.ucar.edu/community-projects/lens/) and CMIP6 data through the Earth System Grid Federation (https://wcrp-cmip.org/cmip-data-access/#access-routes). Radiative kernels and a Python-based
analysis toolkit developed by Ryan Kramer are used to compute the radiative feedbacks discussed in section 5 and are available at https://climate.earth.miami.edu/data/radiative-kernels/index.html. Code for the analysis is available upon request.

## Appendix A: Association of latitude-dependent warming with elevation-dependent warming

Here we assess whether any association of warming with latitude might be able to explain the observed EDW signal, because surface elevation also has a statistical association with latitude. When we average the absolute value of latitude in each bin of
surface elevation, we find that distance from the equator generally increases with surface elevation in our 40°S–40°N domain (Fig. A1a). Using a linear fit to the mean of the absolute latitude within each surface elevation bin, a surface at 5 km elevation lies on average about 14° latitude further poleward than a surface at sea level.

We now decompose the sensitivity of the surface air warming trend ($\delta T_s$) to surface elevation, $d(\delta T_s)/dz_s$, into a sensitivity of warming to latitude, $d(\delta T_s)/d\phi$, and an association of surface elevation with latitude, $d\phi/dz_s$:

$$\frac{d(\delta T_s)}{dz_s} = \frac{d(\delta T_s)}{d\phi} \cdot \frac{d\phi}{dz_s}. \tag{A1}$$

This expression neglects the association of warming with other variables that cannot be expressed in terms of latitude, consistent with our goal of determining whether it alone can explain the observed EDW signal. We estimate the first term within the right-



hand side product from the rate of warming during 1959–2014 of zonal mean land-surface air temperature between the equator and 40°N (Fig. A1b). A linear fit to this rate of warming yields $d(\delta T_s)/d\phi \approx 1.3 \times 10^{-3}$ K/decade/°latitude for ERA5,
with a similar value for HadCRUT5. Combining this sensitivity of warming to latitude with the value of $d\phi/dz_s$ obtained from Fig. A1a, we obtain $d(\delta T_s)/dz_s \simeq 0.0035$ K/decade/km, a value that is about 40% of the ERA5 EDW index of 0.0089 K/decade/km (cf. Fig. 1b).

This relatively small magnitude of $d(\delta T_s)/dz_s$ obtained from (A1) confirms that the majority of the observed EDW signal does not result from the meridional location of orography combined with a general dependence of warming on latitude. Because
of the location of orography at higher latitudes within our 40°S–40°N domain, any positive EDW will contribute to a positive value of $d(\delta T_s)/d\phi$; we thus view the value of $d(\delta T_s)/dz_s$ computed from (A1) as an upper bound on the estimate of the contribution of polar amplified warming to EDW.

*Author contributions.*  All authors contributed to the framing of the paper, to the analyses performed, to discussing the results, and to writing the manuscript.

*Competing interests.*  There are no competing interests.

*Acknowledgements.*  We thank Jonah Bloch-Johnson, Malte Jansen, Tim Merlis, and Nick Pepin for helpful discussions.



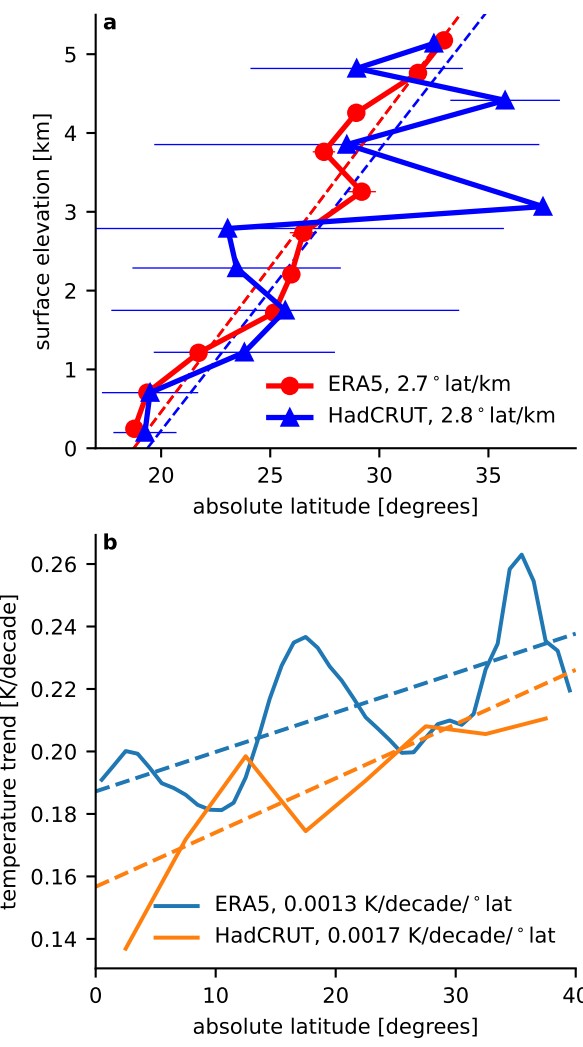

**Figure A1.** (a) Absolute value of latitude (in degrees) binned by surface elevation for the ERA5 reanalysis and HadCRUT5 dataset, plotted as a function of the mean surface elevation in each bin. Error bars represent the 95% confidence interval of the mean in each bin, and linear fits are shown as dashed lines. (b) Linear temporal trend (during 1959–2014, in $\mathrm{K/decade}$) in zonal-mean, annual-mean land surface-air temperature for ERA5 and HadCRUT5 (solid lines), and the linear fits to these between $0°$ and $40°\mathrm{N}$ (dashed lines). Slopes of all linear fits are provided in the legend.



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
