# Peer review of "Elevation-dependent warming: observations, models, and energetic mechanisms"

_EGUsphere, 2024_

## Author Comment (AC1)

**Responses to Reviewer 1's comments**

Link to review: https://doi.org/10.5194/egusphere-2024-31-RC1

We thank the reviewer for their constructive comments and feedback. Below we respond to each point raised:

```
Summary:
The submitted manuscript examines elevation dependent warming (EDW) over
the global tropics and subtropics, using a combination of gridded analyses,
reanalyses, and global climate model simulations. With these datasets,
the authors characterize the magnitude of EDW, the relative roles of
forced response and internal variability in EDW, and the roles of various
forcings and feedbacks in shaping EDW. The authors do a great job of
focusing in on fundamental aspects of EDW and addressing them with
thoughtful, clear, concise, and theoretically sound analyses. The
addressing of forced response vs. internal variability and the formal
diagnosis of mechanisms in a forcing/feedback framework are novel and
very valuable additions to the EDW literature. The prose is crisp and
easy to follow. The figures nicely summarize the results. The authors
leave some questions unanswered (e.g., factors governing the role of
cloud feedbacks in EDW, the role of unresolved terrain) but those
omissions are reasonable given the scope of the study and the
limitations of the datasets analyzed. I only have some small points
for the authors to consider before the manuscript should be ready
for publication.
```

We appreciate the reviewer's positive evaluation of the manuscript.

```
General comments:
I think it is OK that the authors choose to take a global perspective
and focus on global datasets that facilitate their analyses. However,
I think that the authors should at least briefly acknowledge/discuss
what might be missed in such a framework. Regions of interest for EDW
are often in complex terrain that is not sampled well by observation
networks used in historical (re)analyses. Furthermore, the terrain
features associated with EDW may be poorly resolved in GCMs, which
can bias the representation of key processes like accumulation of
mountain snow cover (relevant to the snow albedo feedback) and the
orographic clouds (relevant to cloud feedbacks). How much do the
authors think these factors might affect their results?
```

We agree with the reviewer that the large-scale approach taken in our study has limitations including, in regions of complex terrain, the likely biased representation in global models of important processes (e.g., orographic clouds) and an insufficient observational network for sampling the EDW signal. As suggested, a discussion of these limitations has been added to the revised manuscript (see Lines 359-365 of the revised manuscript) and reads as follows: *"However the large-scale approach taken in this study, using global datasets at relatively coarse spatial resolutions, has limitations. For example, in regions of complex terrain the EDW signal is likely not well sampled by observational networks underpinning the HadCRUT5 and ERA5 datasets. Complex terrain is also an issue for global climate models (e.g., Elvidge et al., 2019) potentially leading to biases in the simulation of processes known to be important for EDW, including mountain snow accumulation and orographic clouds. Future research could investigate the influence of such finer-scale effects on EDW using high-resolution models; work along these lines is already underway (Minder et al., 2018; Palazzi et al., 2019)."*

```
In several places, the authors mention that the surface albedo feedback
is expected to be "stronger at colder temperatures" (or something
similar). I don't think this is quite right. The snow albedo feedback,
which dominates the surface albedo feedback over land, is not
necessarily stronger at cold temperatures. It is often strongest when
near zero deg. C, since small temperature changes can have big effects
on melting and rain vs. snow. I suggest the authors tweak their
wording to reflect this.
```

We thank the reviewer for highlighting this point and agree that the surface albedo feedback is likely to be stronger close to the freezing point rather than at "colder temperatures" as suggested in the original manuscript. We have modified our discussion of this feedback in the revised manuscript to clarify this point (see Lines 98-100 and Lines 29-31).

```
Specific comments:
Introduction: Though it is a few years old, the Pepin et al. (2015) EDW
review paper still seems to be the best overall review on the topic.
Though they site this eventually, I think the authors should cite it
more prominently in the general intro as well.
```

Following the reviewer's suggestion we have added to the introduction two additional citations to Pepin et al. (2015), including to the first sentence of the main text.

L 28{29: might rephrase this to focus on regions where snow/ice is
plentiful, but near freezing point of water (very high/cold locations
may be insensitive to the SAF, due to persistently below-freezing
condition in current and perturbed  climate states). See general
comment #2.

This sentence has been rephrased to emphasise that the surface albedo feedback is
expected to be particularly influential in regions with plentiful snow and ice close to
the freezing point (see Lines 29-31).

L 35{36: Some studies have attempted to quantify and compare the role
of different physical mechanisms in producing EDW, though perhaps in
a less global and/or comprehensive manner as this study. Those
contributions should be more clearly acknowledged here.

To this paragraph (Lines 22-39) we have added four additional citations to better
connect our study to the existing EDW literature (Kotlarski et al., 2012; Pepin et al.,
2015; Palazzi et al., 2017; Minder et al., 2018). We have also added text highlight-
ing the idea that the height-dependence of free-tropospheric warming has been put
forward previously as a potential explanation of EDW (see Line 28). Further, we
have added the following sentence emphasising how the importance of the surface
albedo feedback has been quantified in a study using regional simulations: *"Using
high-resolution simulations, Minder et al. (2018) identified this albedo feedback as the
primary driver of EDW in the Rocky Mountains."* We believe these modifications to
the manuscript help to better connect this study to previous work.

L 50{51: Two potential issues with this dataset should at least be
acknowledged:  1. This relatively coarse resolution is insufficient
to resolve many important topographic features 2. Trends for many
high-elevation grid cells may have large uncertainties, due to a
lack of station locations in mountainous regions.

As discussed in response to a comment above, we have added text to the 'Sum-
mary and conclusions' section highlighting the coarse resolution of the data used in
this study and potential implications (see Lines 359-365). Regarding trends in high-
elevation regions, the large HadCRUT5 error bars in Figure 1 for the high-elevation
bins reflect, in part, the sparse measurements underpinning that dataset in those
regions. The coarse resolution of the HadCRUT5 data and limited spatial coverage
in some regions are also, we believe, well communicated by the middle panel of Fig-
ure 2. We have added the following sentence to the revised manuscript to highlight
limitations with the HadCRUT5 dataset (see Lines 55-56): *"Note that HadCRUT5*

*does not provide complete spatial and temporal coverage due to limited station data in specific regions and at specific times."*

L 50: "observational side...": I suggest adjusting this wording when writing about ERA5, since it is not purely observation-based.

Given ERA5 is an observationally-constrained dataset, we believe the description in the text is reasonable. We have added the word "estimates" to this description to flag that ERA5 does not provide direct measurements (see Line 54).

L 53{55: What is the grid spacing for the CESM1-LE runs?

The resolution of the CESM1-LE simulations is $1° \times 1°$, and this information has been added to the revised manuscript (see Line 58).

L 57: To what extent might the relatively coarse resolution of the models used (as compared to relevant terrain features) bias the simulated EDW?

As outlined above, we have added text to the final section of the manuscript discussing potential implications on simulated EDW of the coarse resolution of global climate models (see Lines 362-365).

In addition, using ERA5, we have investigated the impact on EDW of coarsening the data through horizontal averaging (Figure R1). The differences in trends between the coarsened (cyan line) and uncoarsened data (red line) are minor, suggesting that horizontal resolution does not have a substantial influence on the EDW signal averaged over the tropics and subtropics. We have added the following sentence to the caption of Figure 1 in the main text highlighting that resolution does not have a major influence on the EDW signal: *"Note that quantitatively similar results for ERA5 are obtained when the data are coarsened by horizontal averaging to a resolution of $1° \times 1°$ (not shown)."*

L 93{95: See general comment #2

As discussed above, this section of text has been modified following the reviewer's suggestion (see Lines 98-100).

Section 4: This section provides a nice perspective that is often missing from the EDW literature

Section 5.3: This analysis is great. It concisely breaks down the relevant physical processes without falling into some of the analysis pitfalls of many previous studies that take a more statistical approach.

[Figure]

Figure R1: As in Figure 1a from the main text, but here including an additional (cyan) line showing the ERA5 warming trends binned by elevation using data that have been coarsened by horizontal averaging from $0.25° \times 0.25°$ resolution to $1° \times 1°$ resolution.

We thank the reviewer for the positive comments on Sections 4 and 5.3 of the text.

L 259{260: See general comment #2

The word "colder" has been removed from this sentence to make clear that the surface albedo feedback is not necessarily stronger when conditions are colder, rather it is expected to be more influential when surface snow and ice are prevalent.

L 273{276: One key aspect that is worth mentioning is the strong influence
that mountains have on modulating clouds. The role of the terrain itself
in modulating the cloud cover response could be an important part of the
story (e.g., https://doi.org/10.1175/JCLI-D-21-0379.1). Also, some of
these orographic effects on clouds may not be well resolved in the
relatively coarse resolution simulations considered here.

We thank the reviewer for raising this important point. The sentence on Lines 281-282 of the revised manuscript has been modified to include reference to the modulating

influence of terrain on cloud radiative effects in elevated regions. In this section on cloud feedbacks we have also added a citation to the paper mentioned by the reviewer which analyses the effects of clouds on regional EDW in the Andes (Chimborazo et al., 2022). Finally, we have added text to the final section of the manuscript highlighting potential biases in the simulation of orographic clouds associated with complex terrain (see Lines 362-364).

Section 6: Given the authors' focus on the tropics/subtropics, it would be nice to include a few sentences of discussion to place their results more specifically in the context of previous EDW work that focuses on that part of the world.

To better connect our study to previous work on tropical EDW, we have added a new sentence and new citations (Vuille et al., 2003; Chimborazo et al., 2022) to the revised manuscript (see Lines 102-104). We have added further citations to the Chimborazo et al. (2022) study of the tropical Andes when discussing the impacts of surface albedo (Line 261) and cloud feedbacks (Line 272) on tropical/subtropical EDW.

**References**

Chimborazo, O., J. R. Minder, and M. Vuille, 2022: Observations and simulated mechanisms of elevation-dependent warming over the tropical andes. *J. Climate*, **35 (3)**, 1021–1044, doi:10.1175/JCLI-D-21-0379.1.

Elvidge, A. D., and Coauthors, 2019: Uncertainty in the representation of orography in weather and climate models and implications for parameterized drag. *J. Adv. Model. Earth Sy.*, **11 (8)**, 2567–2585, doi:10.1029/2019MS001661.

Kotlarski, S., T. Bosshard, D. Lüthi, P. Pall, and C. Schär, 2012: Elevation gradients of european climate change in the regional climate model cosmo-clm. *Climatic Change*, **112**, 189–215, doi:10.1007/s10584-011-0195-5.

Minder, J. R., T. W. Letcher, and C. Liu, 2018: The character and causes of elevation-dependent warming in high-resolution simulations of rocky mountain climate change. *J. Climate*, **31 (6)**, 2093–2113, doi:10.1175/JCLI-D-17-0321.1.

Palazzi, E., L. Filippi, and J. von Hardenberg, 2017: Insights into elevation-dependent warming in the tibetan plateau-himalayas from cmip5 model simulations. *Clim. Dynam.*, **48 (11-12)**, 3991–4008, doi:10.1007/s00382-016-3316-z.

Palazzi, E., L. Mortarini, S. Terzago, and J. Von Hardenberg, 2019: Elevation-dependent warming in global climate model simulations at high spatial resolution. *Clim. Dynam.*, **52 (5-6)**, 2685–2702, doi:10.1007/s00382-018-4287-z.

Pepin, N., and Coauthors, 2015: Elevation-dependent warming in mountain regions of the world. *Nat. Clim. Change*, **5 (5)**, 424–430, doi:10.1038/nclimate2563.

Vuille, M., R. S. Bradley, M. Werner, and F. Keimig, 2003: 20th century climate change in the tropical andes: observations and model results. *Climatic Change*, **59**, 75–99, doi:10.1007/978-94-015-1252-7_5.

---

## Author Comment (AC2)

**Responses to Reviewer 2's comments**

Link to review: https://doi.org/10.5194/egusphere-2024-31-RC2

We thank the reviewer for their constructive comments and feedback. Below we respond to each point raised:

```
The authors use the forcing-feedback framework and a feedback analysis
based on radiative kernels to examine the mechanisms causing amplified
warming at higher elevations in low latitudes. The science is strong,
and the manuscript is clear.
```

We appreciate the reviewer's positive evaluation of the manuscript.

```
I have two major and a few minor comments:

I see some similarity between the elevation-dependent warming described
here and the overall tendency for amplified warming in the upper Tropical
troposphere compared to the surface. The latter is caused by the fact
that moist deep convection keeps the Tropical lapse rate close to the
moist adiabat, which is steeper in warmer climates. This is somewhat
akin to the finding that MSE convergence drives EDW, and both are
connected to the increase in water vapour in a warming climate following
Clausius-Clapeyron.
```

We agree with the reviewer that there are qualitative similarities between elevation-dependent warming (EDW), i.e. along-slope warming of near-surface air, and amplified warming in the tropical upper troposphere. We have conducted preliminary analyses of the along-slope and free-tropospheric warming rates, finding that the along-slope trends vary more strongly with height. This differential warming of along-slope versus free-tropospheric air has been noted previously (e.g., Pepin and Seidel, 2005) but is not well understood; we have added a sentence to the revised manuscript highlighting this as a key question for future research (see Lines 366-368).

We also agree with the reviewer that there is likely a connection between EDW and moist convective adjustment, which constrains the vertical profile of moist static energy (MSE) and underpins understanding of amplified warming in the tropical upper troposphere. This is a question we are currently working on and initial results suggest that a theory based on the principle of convective quasi-equilibrium (Emanuel et al., 1994) predicts the correct sign for EDW (i.e., amplified surface air warming at elevation) but overestimates the strength of the historical EDW signal, for reasons that are being investigated. Our aim is to continue this work and write a follow-up paper interpreting EDW using a convective framework.

The authors derive forcing as a residual – could you use the CO2 kernel
instead, possibly with more idealized runs? Deriving forcing as a
residual means there is no residual error to check the quality of the
decomposition, which is unfortunate.

As suggested by the reviewer, a $CO_2$ radiative kernel could be used to diagnose the influence of changing carbon dioxide concentrations on EDW over the historical period. However we are interested in the net effect of radiative forcing on EDW, and $CO_2$ is just one of the forcings influencing temperature over this period (anthropogenic aerosols, volcanic eruptions, and others are also important). For this reason, we decided to use the residual method to estimate the total radiative forcing in the CMIP6 historical simulations. A set of idealised, single-forcing simulations—and kernels for each forcing component—could in principle be used to directly estimate the influences of different forcing agents on EDW over the historical period; this would be an interesting topic for future work.

As discussed in the manuscript (see Lines 296-299), to check the accuracy of the residual method for diagnosing radiative forcing, for a single model (GFDL-CM4) we use the corresponding fixed-SST historical simulation from the Radiative Forcing Model Intercomparison Project (RFMIP; Pincus et al., 2016) to estimate the "effective radiative forcing" (ERF). Comparing the influence of this ERF on temperature trends with the influence of the "instantaneous radiative forcing" (IRF), estimated from the coupled historical simulations using the residual method, we find both methods give similar results (see Figure S1 in the supplement). The sum of the contributions to the temperature trends, when the forcing contribution is estimated using the ERF method rather than as a residual, is also similar to the simulated trends (see Figure R1). This similarity between methods, for a single model, underpins our confidence in using the residual method across the other CMIP6 models (note that many of the models analysed do not have corresponding RFMIP simulations).

Minor comments:
l7: For people who do not know the topic, the sign of EDW (larger
warming at high elevation) is first mentioned halfway through the
abstract, and it is not quite clear if this is a new finding or
corresponds to what was known before.

The tendency for stronger warming over elevated surfaces has been discussed extensively in the literature (e.g., Pepin et al., 2015), but this is the first study (to our knowledge) that analyses the EDW signal on large scales, in different seasons, and across a range of models and observational datasets. To better highlight this distinctive contribution relative to previous work, we have modified the first sentence of

[Figure]

Figure R1: Simulated (solid black line) surface air temperature trends binned by elevation for the GFDL-CM4 historical simulation (1959–2014). The dashed black line shows the sum of the contributions to the temperature trends [see equation (7) in the main text], with the forcing contribution estimated using an "effective radiative forcing" estimated using the RFMIP fixed-SST historical simulation (named *piClim-histall*).

the second paragraph of the abstract so that it now reads (see Lines 5-6): *"Here we expand on previous regional studies and use gridded observations, atmospheric reanalysis, and a range of climate model simulations to investigate EDW over the historical period across the tropics and subtropics (40°S to 40°N)."*

Fig. 1: Why use standard seasons when the data spans the equator? Wouldn't local summer/winter be more consistent?

This is a good suggestion; we have re-plotted Figure 1b in the main text using local seasons rather than standard seasons. The difference is minimal, but using local seasons does increase the EDW index in winter.

If you insist of changing the axis scaling within a Figure, please make a clear visual mark of that.

We have added horizontal dashed lines to Figure 1 in the main text to identify the boundary between the linear and logarithmic regions of the scale (and have noted

this in the revised figure caption). We have also expanded the range of the linear region so that only the CMIP6 outliers fall within the nonlinear region. We hope this makes the figure more acceptable to the reviewer and understandable to readers; we do not insist on this particular scaling, but find it to be a helpful way to see the central estimates and the outliers all on the same plot.

`Fig. 2 What are the gray lines?`

The grey contours mark the 1 km and 2 km surface heights, and thus outline the areas of high orography. Thank you for alerting us to this omission. We have modified the caption of Figure 2 to state this.

`Footnote p 15 - could you include this in the main text?`

Following the reviewer's suggestion, this footnote has been incorporated into the main text (see Lines 296-299).

`l. 316 - see comment above, CC-relation plays a role in MSE gradients`

We agree with the reviewer that CC-mediated changes in surface air specific humidity (along with temperature) undoubtedly influence changing patterns of MSE convergence. We have modified the final sentence of section 5.3.8 to highlight this important point (see Lines 326-327).

**References**

Emanuel, K. A., J. David Neelin, and C. S. Bretherton, 1994: On large-scale circulations in convecting atmospheres. *Quarterly Journal of the Royal Meteorological Society*, **120 (519)**, 1111–1143.

Pepin, N., and D. J. Seidel, 2005: A global comparison of surface and free-air temperatures at high elevations. *J. Geophys. Res.-Atmos.*, **110 (D3)**, doi:10.1029/2004JD005047.

Pepin, N., and Coauthors, 2015: Elevation-dependent warming in mountain regions of the world. *Nat. Clim. Change*, **5 (5)**, 424–430, doi:10.1038/nclimate2563.

Pincus, R., P. M. Forster, and B. Stevens, 2016: The radiative forcing model intercomparison project (rfmip): experimental protocol for cmip6. *Geosci. Model Dev.*, **9 (9)**, 3447–3460, doi:10.5194/gmd-9-3447-2016.